# Association between Immunosenescence, Mitochondrial Dysfunction and Frailty Syndrome in Older Adults

**DOI:** 10.3390/cells12010044

**Published:** 2022-12-22

**Authors:** Ilaria Buondonno, Francesca Sassi, Francesco Cattaneo, Patrizia D’Amelio

**Affiliations:** 1Geriatric and Bone Disease Unit, Department of Medical Science, University of Torino, 10126 Torino, Italy; 2Department of Public Health Sciences and Pediatrics, University of Torino, 10126 Torino, Italy; 3Department of Medicine, Service of Geriatric Medicine & Geriatric Rehabilitation, University of Lausanne Hospital (CHUV), 1011 Lausanne, Switzerland

**Keywords:** frailty, aging, T cells, immunosenescence, cytokine, mitochondria, inflammaging

## Abstract

Aging is associated with changes in the immune system, increased inflammation and mitochondrial dysfunction. The relationship between these phenomena and the clinical phenotype of frailty is unclear. Here, we evaluated the immune phenotypes, T cell functions and mitochondrial functions of immune cells in frail and robust older subjects. We enrolled 20 frail subjects age- and gender-matched with 20 robust controls, and T cell phenotype, response to immune stimulation, cytokine production and immune cell mitochondrial function were assessed. Our results showed that numbers of CD4+ and CD8+ T cells were decreased in frail subjects, without impairment to their ratios. Memory and naïve T cells were not significantly affected by frailty, whereas the expression of CD28 but not that of ICOS was decreased in T cells from frail subjects. T cells from robust subjects produced more IL-17 after CD28 stimulation. Levels of serum cytokines were similar in frail subjects and controls. Mitochondrial bioenergetics and ATP levels were significantly lower in immune cells from frail subjects. In conclusion, we suggest that changes in T cell profiles are associated with aging rather than with frailty syndrome; however, changes in T cell response to immune stimuli and reduced mitochondrial activity in immune cells may be considered hallmarks of frailty.

## 1. Introduction

Aging affects cells, tissues and organs; clinically, we can distinguish between robust and frail aging. Frailty is a complex geriatric syndrome leading to reduced ability of the organism to fight against detrimental factors and, consequently, is associated with decreased functional abilities, dependency, comorbidity and mortality [1]. For these reasons, frailty is associated with a significant increase in health-related costs [2]. At the cellular level, unhealthy aging corresponds to senescence, which has been suggested to play a causal role in the development of the frailty syndrome [3].

Several biological mechanisms account for unhealthy aging and the development of frailty; amongst these is the deregulation of the immune system, which reduces the ability to fight against microorganisms and cancer invasion, known as “immunosenescence” [4,5]. The immune system significantly changes with aging; in particular, a general reduction in naïve T lymphocytes and an increase in memory T cells has been observed in older adults [6,7]. Moreover, an imbalance in T lymphocyte sub-sets with changes in the ratios of T helper (CD4+) cells to cytotoxic T (CD8+) cells have been reported [8,9]. Besides changes in the numbers and ratios of different T cells, a reduction in the levels of the co-stimulatory receptor CD28 has been described [10,11]. On the other hand, in mouse models, levels of the inducible T cell co-stimulator (ICOS), which mainly inhibits T cell function [12], were found to be increased [13]. The increased levels of inhibitory co-receptors in aging mouse T cells, together with the reduction in stimulatory molecules such as CD28, suggests a reduction in T cell response to immune stimuli which may lead to immune system dysfunction, decreased defense against infections and cancer, and increased susceptibility to autoimmune diseases in older age. The decrease in naïve T cells during aging may lead to the reduced abilities of older patients to respond to unknown immune stimuli, leading to increased susceptibility to infections and cancer and reduced immune response to vaccines [14,15,16]. The impairment of mitochondrial function associated with aging [17] has been proposed as one of the main contributors to the development of immunosenescence and immune system dysfunction [18,19,20]. Immunosenescence and immune cell mitochondrial dysfunction may be the cause of the increase in sterile inflammation associated with aging named “inflammaging” [21,22].

Based on these findings, some authors have suggested that it is possible to identify an immunological risk phenotype associated with increased mortality in older adults [23,24,25], suggesting that immunosenescence and inflammaging correspond to a clinical phenotype of unsuccessful aging. However, despite these interesting suggestions, claims about the effects of immunosenescence on immune function and clinical outcomes have been challenged in recent years. In particular, it has been shown that the reduction in naïve T cells can be counteracted by their proliferation after IL-7 stimulation and by the recently described stem-cell-like memory T cells, which may replenish the naïve T cell compartment when needed during aging [26]. Thanks to these homeostatic mechanisms, even in older age, T cells may be able to maintain acceptable immune functions [4]: according to these data, immunosenescence may not play a role in determining clinical impairment of the immune system and hence in the development of frailty.

The present study aims to unravel the association between immunosenescence, mitochondrial dysfunction and frailty syndrome in older community-dwelling frail adult subjects age- and gender-matched with robust subjects.

## 2. Materials and Methods

### 2.1. Patients and Clinical Evaluation

The protocol for this case–control study was approved by the Ethical Committee of the “Città della Salute e della Scienza (Torino)” on 23 May 2018, protocol number 0053124. All the enrolled subjects signed an informed consent form before their enrolment. The patients included in the present study belong to a larger cohort that has already been described [27]. Twenty frail older adults and 20 age- and gender-matched robust controls were recruited amongst patients who went to the outpatient service of the Geriatric Unit of the University of Torino (Italy) or to the general practitioner. In order to avoid selection biases, patients suffering from immune disorders, dementia, cancer and renal failure were excluded from the study. Patients taking drugs acting on the immune system, such as corticosteroids and immunosuppressants, were also excluded from the study. All patients underwent a comprehensive geriatric assessment in order to determine their functional, mental and nutritional statuses. The clinical evaluation of the patients has been fully described elsewhere [27]. Patients were classified as frail or robust according to the Fried criteria [27,28].

### 2.2. T Cell Activation

Peripheral blood mononuclear cells (PBMCs) were obtained via the Ficoll–Paque technique, as previously described [29]. A quantity of 1.5 × 10^6^ PBMCs were plated on 96-well plates and activated after overnight incubation at 4 °C with anti-CD3 Ab (OKT3, 1 μg/mL, eBioscience, San Diego, CA, USA). After overnight incubations, to stimulate ICOS or CD28, the plates were washed with PBS and further coated with B7h-Fc (5 μg/mL; Bio-Techne, Minneapolis, MN, USA) or B7.1-Fc (5 μg/mL; Bio-Techne, Minneapolis, MN, USA) for 2 h at room temperature. The plates were then washed with PBS, and PBMCs were cultured in RPMI 1640 (Invitrogen, Burlington, ON, Canada) plus 10% FBS (Invitrogen, Burlington, ON, Canada) for 5 days, after which the PBMCs were collected and analyzed via flow cytometry.

### 2.3. Flow Cytometry

T cell phenotypes were analyzed at baseline by flow cytometry after staining with CD4-FITC or CD8-FITC, CD45RO-PE, CD45RA-PERCP, ICOS-APC or CD28-APC antibodies or with the corresponding isotype controls, according to the manufacturer’s instructions (antibodies obtained from Biolegend, San Diego, CA, USA). Flow cytometry was performed with an FACS Calibur flow cytometer and Cell Quest Software (BD Biosciences, Franklin Lakes, NJ, USA), and each analysis consisted of at least 100,000 events recorded within the lymphocyte gate. In order to evaluate the ability of T cells to respond to immune stimulation, the cells were analyzed at baseline and after immune stimulation with B7.1-Fc or with B7h-Fc.

### 2.4. Cytokine Production

In order to measure the systemic effects of T cell activity and inflammation, we measured the levels of serum IL17, IL4, TNFα, IFNγ and ICOSL by the ELISA technique (R&D Systems^®^, Minneapolis, MN, USA). The production of cytokines by PBMCs stimulated with B7.1-Fc or B7h-Fc was evaluated by the ELISA technique in cellular supernatants. The levels of IL17, IL4, TNFα and IFNγ (R&D Systems^®^, Minneapolis, MN, USA) were measured. All the experiments were performed in duplicate.

### 2.5. Functional Evaluation of Mitochondrial Activity

Mitochondrial fractions were isolated from PBMCs according to a standard technique described in detail elsewhere [30]. Briefly, PBMCs were lysed in 0.5 mL buffer A (50 mMTris, 100 mMKCl, 5 mM MgCl2, 1.8 mM ATP, 1 mM EDTA, pH 7.2) supplemented with protease inhibitor cocktail III (Calbiochem), 1 mM PMSF and 250 mMNaF and mitochondrial fraction obtained after centrifugation. The activities of complexes I–III were measured in 10 µL of non-sonicated mitochondrial samples suspended in buffer C (5 mM KH2PO4, 5 mM MgCl2, 5% *w/v* BSA) and transferred into a quartz spectrophotometer cuvette. Then, buffer D (25% *w/v* saponin, 50 mM KH2PO4, 5 mM MgCl2, 5% *w/v* BSA, 0.12 mM cytochrome c-oxidized form, 0.2 mM NaN3) was added for 5 min at room temperature. The reaction was started with 0.15 mM NADH and was followed for 5 min, and the absorbance was read at 550 nm with a Lambda 3 spectrophotometer (PerkinElmer). In these experimental conditions, the rate of cytochrome c reduction, expressed as nmol cyt C reduced·min−1·(mg cell protein)−1, was dependent on the activity of both complex I and complex III. Thereafter, 0.05 mM of the complex I inhibitor rotenone was added to the cuvette, and the reaction was followed for a further 5 min. In this phase, the rate of cytochrome C reduction was dependent on ubiquinone and complex III activity only and was taken as an index of the amount of active ubiquinone in mitochondria; in the presence of rotenone, the electron flux is reduced to less than 5%. Results were expressed as nmol reduced cytochrome C/min/mg mitochondrial proteins. The described procedure is routinely used in our laboratory [30].

Amounts of ATP were measured in 20 µg of mitochondrial extracts using the ATP Bioluminescent Assay Kit (FL-AA, Sigma Aldrich Co., St. Louis, MO, USA) and a previously set calibration curve. Data are expressed as nmol/mg mitochondrial proteins.

### 2.6. Statistical Analyses

Frail and robust subjects were compared with respect to the analyzed variables by means of ANOVA one-way tests for continuous Gaussian variables and the Mann–Whitney U test for continuous non-Gaussian variables. Comparisons between baseline and post-stimuli variables were performed by 2-way ANOVA for repeated measures; multiple comparisons were performed by means of Šídák’s multiple comparisons test.

Mitochondrial ATP level and electron chain flux activity were correlated with muscle mass and function by Pearson’s coefficient correlation.

SPSS 27.0 was used for the statistical analyses, and *p* < 0.05 was considered statistically significant. Graphs were drawn using GraphPad 8.0 for Windows.

## 3. Results

Subjects defined as frail met at least three Fried criteria [28]: 11 patients displayed four out of five Fried criteria and 4 met all five criteria. Frail subjects had higher comorbidity indexes, even though the number of drugs taken daily was comparable amongst the two groups; they had worse nutritional status; and they were more dependent. As regards cognitive performance, the Mini Mental State Examination (MMSE) [31] was similar between the two groups, whereas mood, evaluated with the Geriatric Depression Scale (GDS) [32], was more depressed in frail as compared with robust subjects. Physical activity was reduced in frail subjects as measured by the 4 m walking test and the Physical Activity Scale for the Elderly (PASE) [33]. Muscle function was significantly impaired in frail subjects, with a reduction in strength as measured by handgrip strength [34] and in performance as measured by the Short Performance Physical Battery (SPPB) [35]; muscle mass was not significantly reduced in frail subjects. Table 1 shows the general characteristics of the two groups.

In frail subjects, both CD4+ and CD8+ T cells were decreased as compared with robust subjects, regardless of whether the ratios of CD4 to CD8 T cells were significantly different. Naïve (CD45Ra+) and memory (CD45Ro+) T cells, as well as T cells expressing ICOS, were similar between the two groups, whereas both CD4 and CD8 T cells expressed lower levels of CD28 in frail as compared with robust subjects (Table 2).

Despite the differences in immune phenotypes, there were no significant differences in the serum levels of the measured cytokines between frail and robust subjects (Table 3).

In order to evaluate the responses to immune stimuli in T cells in frail and robust subjects, we stimulated PBMCs with B7.1-Fc or B7h-Fc antibodies. Despite the reduced expression of CD28 in T cells from frail subjects, T cells from both groups responded similarly to the stimuli (Figure 1).

Stimulation with B7h-Fc was effective in reducing TNFα production in frail and robust patients (Figure 2A), whereas T cells from robust patients produced more IL-17 after B7.1-Fc stimulation as compared with T cells from frail subjects (Figure 2H).

In order to evaluate possible impairment of mitochondrial function in PBMCs from frail and robust subjects, we measured their electron flux chains and ATP levels and found that both parameters were significantly reduced in frail subjects (Figure 3). As mitochondrial function is reduced in sarcopenia, which is a key player in the diagnosis of frailty [27], we checked for correlations between mitochondrial bioenergetics and muscle mass and function. Our data revealed a good correlation between ATP mitochondrial level and ASMM (R = 0.63, *p* = 0.049), 4 m walking speed (R = 0.68, *p* = 0.32) and handgrip strength (R = 0.65, *p* = 0.41); no correlations were found between muscular parameters and electron chain fluxes.

## 4. Discussion

Whether the observed changes in T cell phenotypes in older subjects are responsible for a clinically detectable difference in immune system function and the development of frailty is currently under debate. Clinical studies in centenarians show that these patients do not have increased prevalence of opportunistic infections and/or cancers; on the contrary, longitudinal studies suggest that having lower numbers of CD8+ cells inducing inflammatory responses to cytomegalovirus may be considered a survival advantage in older patients [36,37]. However, there are no specific studies addressing the question of whether the immune phenotype changes observed are associated with impaired response to T cell stimulation and altered mitochondrial function in frailty syndrome. To answer this provocative question, we evaluated the differences in T cell phenotypes, responses to immune stimuli and PBMC mitochondrial function in frail and robust community-dwelling older subjects matched with respect to age and gender.

Our data showed a general reduction in both CD4- and CD8-positive T cells in frail older adults as compared with age-matched controls; however, we did not find a significant change in their ratios in frail subjects. Previous studies suggested that a CD4/CD8 ratio lower than 1 may be considered a risk for poor health outcomes [38,39]; however, the results for the different cohorts were not all similar [11,40,41]. A recent paper on a very large cohort of subjects suggested that the immune phenotype observed in older age depends on the accumulation of several acute or chronic stressors and that lifestyle factors and infection by cytomegalovirus may partially counteract these effects [42]; however, the authors did not analyze a possible role for the frailty syndrome in this context.

According to our findings, the imbalance between CD4+ and CD8+ cells may not be regarded as a marker of frail aging, thus confirming the hypothesis raised in clinical longitudinal studies according to which no survival advantage is associated with increased CD4/CD8+ T cell ratios [36,37], while reductions in both CD4 and CD8 positive T cell numbers may be associated with frail aging.

Previous studies showed a decrease in naïve T cells in older age [14,15,16], and here we have shown that in frail subjects there was no significant imbalance between naïve and memory CD4+ and CD8+ T cells as compared with robust subjects, suggesting that decrease in naïve T cells is not associated with frailty. In this regard, a cross-sectional study on nursing-home residents showed that lower naïve CD4+ T cell and memory CD8+ T cell numbers are associated with higher frailty indexes; however, as in our study, these authors did not find an association between a CD4+/CD8+ T cell ratio less than 1.0 and frailty syndrome or mortality [43]. Although the cohort examined by Johnston and colleagues [43] was very large, they examined nursing-home residents and not community-dwelling patients and they measured frailty using the frailty index and not the Fried phenotype; the different setting may explain the different results obtained in our study.

Our data showed reduced expression of CD28 on both CD4+ and CD8+ T cells in frail patients compared to robust ones, confirming data obtained in different settings by other researchers [10,11]. CD28 stimulation promotes the proliferation of stimulated T cells and plays a pivotal role in the T cell-dependent antibody responses; hence, its reduction suggests a reduced ability of T cells to respond to T cell receptor stimuli [44]. We also investigated the expression of the inhibitory co-receptor ICOS, which belongs to the CD28 family and has been suggested to increase during aging in mouse models [13]. Here, we have shown that ICOS expression was similar on T cells from frail and robust subjects; to our knowledge, this is the first paper to measure ICOS expression in T cells from frail and robust older subjects.

In order to clarify the functional role of the differential expression of CD28 and ICOS in frailty, we evaluated the functional responses of T cells to stimuli that activate the CD28 and ICOS receptors. We have shown that, despite the differential expression of CD28 on T cells from frail subjects, responses to the immune stimuli were similar in the two subject groups in terms of T cell proliferation; nevertheless, stimulation of CD28 was not able to rescue the numbers of CD4+ and CD8+ T cells in frail subjects. Even though we did not observe a significant difference in T cell proliferation after stimulation with B7.1-Fc between frail and robust subjects, the functional effect on cytokine production was significantly different in frailty syndrome. Frail subjects produced less IL-17 after stimulation of CD28 as compared with robust ones. We did not observe significant differences between the two groups after stimulation of ICOS. Despite the differences observed in vitro, we did not find any significant differences in serum cytokine levels between frail and robust subjects. Some pro-inflammatory cytokines, such as IL-6, have been shown to be increased in frailty, both in humans and in mice, by a rather large amount of evidence. (For a complete review, see Heinze-Milne and colleagues [45]). However, data on the levels of other cytokines in frailty are less certain. As regards the cytokines measured in our study, there are conflicting reports, as some studies have suggested that they are increased while others have suggested that they are not. In particular, contrasting results have been found for IL-17, levels of which were found to be decreased in frail subjects in one study on male patients [46] and unchanged in others [47,48]. As regards levels of IFNϒ, there is some controversy, as one study suggested that this cytokine is increased in frail patients [49], while others have claimed that it is not [46,47,48]. Controversial results have been obtained, also, for TNFα, as some authors have shown an increased level of this cytokine in frailty [50,51,52,53], while others have shown it not to be increased [46,48,54]. By contrast, there is more agreement on unchanged levels of IL-4 in frailty; even though this cytokine has been measured in only two studies, the two agreed on no significant difference between frail and non-frail subjects [47,48]. To our knowledge, there are no previous data available on the levels of ICOSL in frailty; thus, this is the first study to report results on this subject.

Despite the literature showing that mitochondrial function declines with age and that this may play a role in age-associated increase in inflammation and immunosenescence [55], there are no studies investigating this relation in frailty in humans. Here, we have shown that the mitochondrial function of PBMCs from frail patients was compromised as compared with mitochondrial function in age-matched robust subjects. In our previous work, we demonstrated that mitochondrial function in PBMCs from old malnourished patients can be boosted with targeted nutritional support and that this activity is correlated with improvement in clinical outcomes [30]. Our data suggest that mitochondrial dysfunction may be considered a hallmark of frailty and targeted as a possible therapeutic intervention [19]. Mitochondrial function is correlated with muscular strength, performance and mass, and these are key features of sarcopenia, which is associated with physical frailty [27,56]. Here, we have confirmed previous data suggesting that mitochondrial impairment is a key feature of sarcopenia [19,30].

The observed reduction in CD28 expression on T cell surfaces may play a pivotal role in the impairment of mitochondrial bioenergetics in frail subjects, as Geltink and colleagues [57] demonstrated that CD28 signals are implicated in mitochondrial function. In particular, during T cell activation, the stimulation of CD28 induces the remodeling of mitochondrial cristae and enhances respiratory capability, thus allowing cytokine production by T cells upon re-stimulation.

From a methodological point of view, it is noteworthy that, despite known differences in mitochondrial contents in different PBMC sub-populations [58,59], our data on mitochondrial activity were not biased by the differences observed in T cells from frail and robust subjects, as mitochondrial activity was displayed after normalization for mitochondrial proteins.

The main limitation of our study is the small sample size and the absence of follow-up; it would be interesting to assess whether reduced T cell activation and PBMC mitochondrial dysfunction is associated with negative clinical outcomes, such as loss of independence, muscular strength and function over time. Despite these limits, the subjects enrolled were well characterized from clinical and biological points of view, and the control subjects were age- and gender-matched with frail subjects in order to reduce selection bias. As an additional limitation, we acknowledge that we cannot indicate a clear causal mechanism relating immunosenescence and mitochondrial dysfunction to frailty; however, the data from our study may pave the way for further longitudinal human studies addressing the causal relationship between mitochondrial dysfunction, immunosenescence and the development of frailty syndrome.

## 5. Conclusions

In conclusion, we suggest that changes in T cell profiles are not associated with frailty syndrome; however, changes in the numbers of T cells, their responses to immune stimuli and reduced mitochondrial activity in PBMCs may be considered hallmarks of frailty.

## Figures and Tables

**Figure 1 cells-12-00044-f001:**
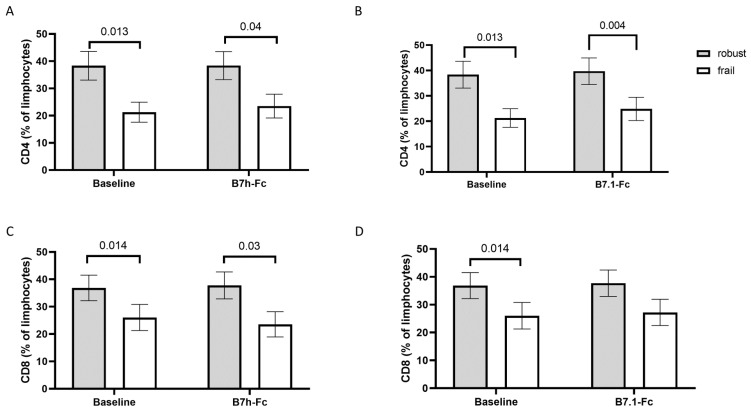
CD4+ and CD8+ T cells in frail and robust older subjects after immune stimulation. Panel (**A**) CD4+ T cells in frail and robust subjects at baseline and after stimulation with B7h-Fc. Panel (**B**) CD4+ T cells in frail and robust subjects at baseline and after stimulation with B7.1-Fc. Panel (**C**) CD8+ T cells in frail and robust subjects at baseline and after stimulation with B7h-Fc. Panel (**D**) CD8+ T cells in frail and robust subjects at baseline and after stimulation with B7.1-Fc. The graphs show means and standard deviations; data were analyzed by two-way ANOVA for repeated measures; multiple comparisons were performed by means of Šídák’s multiple comparisons test. Significant *p*-values are shown.

**Figure 2 cells-12-00044-f002:**
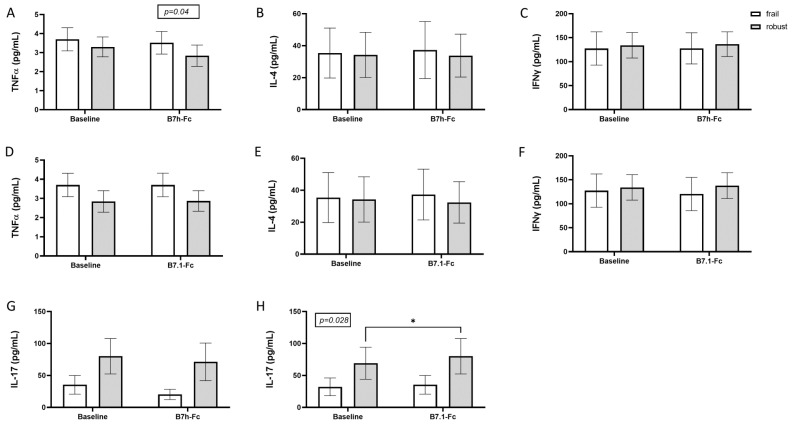
In vitro cytokine production in PBMCs from frail and robust older subjects after immune stimulation. Panel (**A**) TNFα in frail and robust subjects at baseline and after stimulation with B7h-Fc. Panel (**B**) IL-4 in frail and robust subjects at baseline and after stimulation with B7h-Fc. Panel (**C**) IFNϒ cells in frail and robust subjects at baseline and after stimulation with B7h-Fc. Panel (**D**) TNFα in frail and robust subjects at baseline and after stimulation with B7.1-Fc. Panel (**E**) IL-4 in frail and robust subjects at baseline and after stimulation with B7.1-Fc. Panel (**F**) IFNϒ in frail and robust subjects at baseline and after stimulation with B7.1-Fc. Panel (**G**) IL-17 in frail and robust subjects at baseline and after stimulation with B7h-Fc. Panel (**H**) IL-17 in frail and robust subjects at baseline and after stimulation with B7.1-Fc. The graphs show means and standard deviations; data were analyzed by two-way ANOVA for repeated measures; multiple comparisons were performed by means of Šídák’s multiple comparisons test. Significant *p*-values between baseline and after stimuli are indicated as asterisk (*); *p*-values reported in squares are values obtained by two-way ANOVA tests.

**Figure 3 cells-12-00044-f003:**
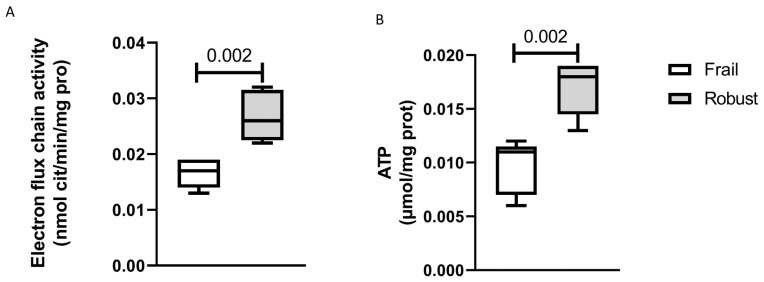
Mitochondrial activity in PBMCs from frail and robust older subjects. Panel (**A**) The box and whisker plot shows electron flux chain activity (on the y-axis) in mitochondria from frail and robust subjects. Panel (**B**) The box and whisker plot shows the ATP levels (on the y-axis) in mitochondria from frail and robust subjects. Box and whiskers are drawn according to the Tukey method; median and interquartile range maximum and minimum values are shown. *p*-values were calculated by one-way ANOVA.

**Table 1 cells-12-00044-t001:** General characteristics of the subjects according to the diagnosis of frailty. Mean and SD values with CIs are shown; *p*-values were calculated by one-way ANOVA.

	Robust (20)	Frail (20)	*p*-Value
Age (years)	83 ± 5	82 ± 6	0.469
Gender (number)	10 W, 10 M	10 W, 10 M	0.752
Number of drugs/daily	4.5 ± 2.4 (3.4–5.6)	5.0 ± 2.3 (3.9–6.0)	0.523
BMI	23.6 ± 2.7 (22.4–24.9)	22.9 ± 4.5 (20.8–25.0)	0.527
MNA (score/30)	23.1 ± 4.5 (21.1–25.2)	20.4 ± 4.1 (18.5–22.3)	0.049
Four-meter walking test (m/sec)	0.84 ± 0.2 (0.8–0.9)	0.5 ± 0.2 (0.5–0.6)	<0.0001
PASE (score)	131.4 ± 25.1 (119.9–142.8)	67.6 ± 27.1 (55.0–90.31)	<0.0001
Handgrip strength (Kg)	28.9 ± 5.9 (26.2–31.6)	17.5 ± 6.1 (14.6–20.3)	<0.0001
ASMM (Kg/m^2^)	7.6 ± 2.7 (6.3–8.8)	8.1 ± 4.3 (6.1–10.12)	0.653
SPPB (score/30)	8.4 ± 1.9 (7.4–9.1)	3.9 ± 3.1 (2.4–5.4)	<0.0001
MMSE (score/30)	27.4 ± 1.3 (26.8–27.9)	26.8 ± 1.3 (26.2–-27.4)	0.140
CIRS (score/30)	7.9 ± 3.5 (6.3–9.4)	11.3 ± 4.7 (9.0–13.5)	0.012
GDS (score/30)	8.4 ± 4.0 (6.6–10.3)	16.1 ± 8.6 (12.4–20.4)	<0.0001
ADL (number of lost functions)	0.5 ± 0.48 (0.05–0.15)	1.1 ± 0.4 (0.2–1.9)	0.013
IADL (score/14)	11.1 ± 3.3 (6.5–11)	8.8 ± 4.8 (6.5–11)	0.08

ASMM = appendicular skeletal muscle mass, measured by bioelectric impedance analyses; women (W), men (M).

**Table 2 cells-12-00044-t002:** T cell phenotypes in frail and robust older subjects. Mean and SD values with CIs are shown for Gaussian variables. Median, 25th and 75th percentiles are shown for non-Gaussian variables, indicated with an asterisk (*). *p*-values were calculated by one-way ANOVA for Gaussian variables and by the Mann-Whitney U test for non-Gaussian variables.

T Cell Type (% of Lymphocytes)	Frail (20)	Robust (20)	*p*-Value
CD8+ T cells	21.5 ± 15.7 (13.7–29.3)	38.8 ± 22.2 (27.4–48.2)	0.014
CD8+CD45 Ro+ T cells *	14.8 (5.7–23.3)	8 (0.9–28.9)	0.602
CD8+ CD45Ra+ T cells *	1.6 (0.6–3.1)	0.6 (0–2.2)	0.211
CD4+ T cells	21.3 ± 15.5 (13.6–29)	38.3 ± 23.6 (27.3–49.4)	0.013
CD4+ CD45Ro+ T cells	12.9 (2.3–19.7)	26.3 (1–28.9)	0.871
CD4+ CD45Ra+ T cells *	1 (1–1.5)	0.4 (0.1–2.3)	0.620
CD4+/CD8+	1 ± 0.1 (1–1.1)	1 ± 0.1 (0.9–1.1)	0.781
C8+/ICOS+	35.5 ± 5.4 (24.2–46.9)	26.8 ± (17.2–36.3)	0.218
C4+/ICOS+	2.9 ± 0.9 (1.1–-4.7)	2.7 ± 0.7 (1.1–4.2)	0.842
CD8+/CD28+	24.5 ± 4.5 (15–34)	38 ± 4.8 (27.9–48.2)	0.048
CD4+/CD28+	24.9 ± 4.6 (15.2–35.5)	39.7 ± 5.2 (28.8–50.6)	0.041

**Table 3 cells-12-00044-t003:** Serum cytokines in frail and robust older subjects. Mean and SD values with CIs are shown. *p*-values were calculated by one-way ANOVA.

Serum Cytokines	Frail (20)	Robust (20)	*p*-Value
IL17 (pg/mL)	268.3 ± 38.3 (190.6–346.1)	378 ± 52 (273.6–482.5)	0.098
IFNγ (pg/mL)	284.0 ± 13.9 (255.8–312.3)	309.1 ± 14.7 (279.3–338.9)	0.222
IL4 (pg/mL)	580.7 ± 43.8 (491.9–669.5)	526.4 ± 38.6 (448.4–604.4)	0.354
TNFα (pg/mL)	283.2 ± 18.6 (245.4–321.0)	306.3 ± 22.2 (261.3–351.2)	0.434
ICOSL (ng/mL)	11.4 − 1.6 (8.2–14.5)	11.4 ± 1.6 (8.3–14.5)	0.981

## Data Availability

Data are available upon reasonable request from the corresponding author.

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
