# Peer review of "Association between Immunosenescence, Mitochondrial Dysfunction and Frailty Syndrome in Older Adults"

_cells, 2022, doi:10.3390/cells12010044_

Round 1
Reviewer 1 Report
BUONDONNO et al. try to address whether frailty during aging is associated with immune phenotype, T cell function, and mitochondrial function.
- The title is misleading since the study is only descriptive in nature and therefore in no way answer the question.
- Patient informed consents is not reported.
- Table 1 should display patients characteristics.
- Authors mention they report ATP production in fig 3. I believe it is rather ATP level.
- Rotenone is used as a negative control for the complex I-III activity assay. Can the authors show the data? in the presence of rotenone, activity should reflex the rate of cytochrome c reduction was dependent ubiquinone and Complex III activity.
- The authors cannot conclude their data support T cell profile is associated with aging since their report only describe aged patients.
- Can the authors discuss the fact that mitochondrial differences in complex I/III activity and ATP levels may only reflect differences in PBMC cell composition?
- English grammar should be checked
Author Response
Reviewer: BUONDONNO et al. try to address whether frailty during aging is associated with immune phenotype, T cell function, and mitochondrial function.
The title is misleading since the study is only descriptive in nature and therefore in no way answer the question.
Authors: the reviewer is most gratefully thanked for the useful comments. The title has been changed according to the reviewer’s suggestion
Reviewer: Patient informed consents is not reported.
Authors: information on informed consent has been added
Reviewer: Table 1 should display patients characteristics.
Authors: table 1 with patients characteristics has been added
Reviewer: Authors mention they report ATP production in fig 3. I believe it is rather ATP level.
Authors: the reviewer is right, corrected, thank you
Reviewer: Rotenone is used as a negative control for the complex I-III activity assay. Can the authors show the data? in the presence of rotenone, activity should reflex the rate of cytochrome c reduction was dependent ubiquinone and Complex III activity.
Authors: we use rotenone as negative control, the activity measured after the addiction of rotenone is reduced to less than 5% according to a standard technique. This is now better explained in the materials and methods section.
Reviewer:The authors cannot conclude their data support T cell profile is associated with aging since their report only describe aged patients.
Authors: reworded
Reviewer: Can the authors discuss the fact that mitochondrial differences in complex I/III activity and ATP levels may only reflect differences in PBMC cell composition?
Authors: data for mitochondrial activity are expressed after normalization for mitochondrial proteins, so it appears unlikely that our data are influenced by differences in PBMC subpopulations between frail and controls, a paragraph on this topic was added to the discussion as requested by the reviewer.
Reviewer: English grammar should be checked
Authors: done
Reviewer 2 Report
Here the authors compare the T cell profile of elderly robust patients vs frial patients. The results are really interesting, connecting the phenotypes observed with specific cellular metabolic features.
Some of the claims should be better discussed, for example when they suggest that the changes observed are rather associated to aging than frailty syndrome.
The phenotypes observed on CD28 and mitochondrial performance are really interesting. Authors should reference this paper Geltink R. et al Cell 2017, associating CD28 to mitochondrial priming.
Other minor comments:
-authors in line 166 should also reference Fig2H
-authors should make the colors of the bar graphs consistent throughout the paper
-in Fig.3A, the figure legend should mention what is represented in y axis
Author Response
Reviewer: The phenotypes observed on CD28 and mitochondrial performance are really interesting. Authors should reference this paper Geltink R. et al Cell 2017, associating CD28 to mitochondrial priming.
Authors: the reviewer is most gratefully thanked for the useful comments. We enriched the discussion section and added the paper from Geltink R. et al as suggested
Reviewer: authors in line 166 should also reference Fig2H
Authors: corrected.
Reviewer: authors should make the colors of the bar graphs consistent throughout the paper
Authors: done
Reviewer: in Fig.3A, the figure legend should mention what is represented in y axis
Authors: done
Reviewer 3 Report
The manuscript is interesting, well written and informative. Although small sized, the study contributes to elucidate some important cellular mechanisms of frailty in older people.
I have the following comments:
- There is no clinical characterization of participants. The key clinical characteristics of frail subjects and controls, including of course age, sex, level of motoric and cognitive performance, chronic illnesses, drugs, and so on, should be introduced in a Table and briefly discussed at the beginning of the results section.
- The authors affirm that frailty was diagnosed in accordance with the Fried criteria. I think that the number of criteria fulfilled by each participant should be included in the results. Data on gait speed should be also present.
- Mitochondrial dysfunction is an important characteristic occurring in skeletal muscle cells of subjects suffering from sarcopenia. Sarcopenia frequently overlaps with frailty in older people. It would be interesting to include some data on skeletal muscle mass and function of participants, especially in relation to mitochondrial function.
- The study is limited by the small sample size and by the absence of prospective evaluation of participants. It would have been really interesting to assess whether reduced T cell activation and PBMC mitochondrial dysfunction was associated with increased burden of frailty and loss of independence over time. Maybe a more detailed discussion on study limitations, including small sample size, could be introduced in the manuscript.
Author Response
Reviewer: There is no clinical characterization of participants. The key clinical characteristics of frail subjects and controls, including of course age, sex, level of motoric and cognitive performance, chronic illnesses, drugs, and so on, should be introduced in a Table and briefly discussed at the beginning of the results section.
Authors: the reviewer is most gratefully thanked for the useful comments. The requested table has been added and briefly discussed as requested
Reviewer: The authors affirm that frailty was diagnosed in accordance with the Fried criteria. I think that the number of criteria fulfilled by each participant should be included in the results. Data on gait speed should be also present.
Authors: the requested information have been added in the results section.
Reviewer: Mitochondrial dysfunction is an important characteristic occurring in skeletal muscle cells of subjects suffering from sarcopenia. Sarcopenia frequently overlaps with frailty in older people. It would be interesting to include some data on skeletal muscle mass and function of participants, especially in relation to mitochondrial function.
Authors: the information have been added as requested
Reviewer: The study is limited by the small sample size and by the absence of prospective evaluation of participants. It would have been really interesting to assess whether reduced T cell activation and PBMC mitochondrial dysfunction was associated with increased burden of frailty and loss of independence over time. Maybe a more detailed discussion on study limitations, including small sample size, could be introduced in the manuscript.
Authors: the discussion is now updated according to the reviewer suggestion
Round 2
Reviewer 1 Report
In the title : your study is not case-control study (which is a very specific type of study)
I would suggest: Association of immune-senescence, mitochondrial dysfunction and frailty syndrome in elderly subjects
Table 1 should display age and gender.
Author Response
Reviewer: In the title : your study is not case-control study (which is a very specific type of study)
I would suggest: Association of immune-senescence, mitochondrial dysfunction and frailty syndrome in elderly subjects
Authors: the study has been designed as a case-control study, however we have changed the title according to reviewer’s suggestion, as follows:
“Association between immune-senescence, mitochondrial dysfunction and frailty syndrome in older adults”.
The term “older” has been used instead of “elderly” as it is perceived as less stigmatizing according to the UN Committee on Economic, Social, and Cultural Rights of Older Person (1995) [1]
Reviewer: Table 1 should display age and gender.
Authors: done
References.
- General Comment No. 6: The Economic, Social and Cultural Rights of Older Persons.
Reviewer 3 Report
The authors have satisfactorily responded to all my previous comments. I have no further comments.
Author Response
The manuscript has been revised for spelling errors, thank you.